# Urban Child Labor in Bangladesh: Determinants and Its Possible Impacts on Health and Education

**Md Abdul Ahad** [1,2,*], **Mitu Chowdhury** [1], **Yvonne K. Parry** [2] and **Eileen Willis** [2]

1  Department of Rural Sociology & Development, Sylhet Agricultural University, Sylhet 3100, Bangladesh; mitu_bs@yahoo.com
2  College of Nursing & Health Sciences, Flinders University, Adelaide, SA 5042, Australia; yvonne.parry@flinders.edu.au (Y.K.P.); eileen.willis@flinders.edu.au (E.W.)
*  Correspondence: ahad0005@flinders.edu.au

**Abstract:** (1) Background: A significant proportion of child laborers are compelled to work in exploitative environments, and experience both deteriorating health and financial loss. The present study sought to determine the factors affecting child labor and the characteristics of their working environment. (2) Methods: A questionnaire survey was conducted with 80 child laborers aged 5 to 17 years. Alongside descriptive statistics, a newly devised technique known as the Influencing Causes Index (ICI) was administered and tested. (3) Results: The demographic findings reveal that most child laborers are young children (12–14 years) and 32.5% of child laborers have never attended school. The thorough assessment of determinants reflects that not only poverty but schooling expenses and a lack of access to opportunities in primary schools are also the top-ranked push factors to trigger children towards labor. Around 72.5% of children work for over 8 h a day. A significant proportion of participants received no leave, training, or access to hygiene facilities. The existing pattern of employment and working conditions resulted in musculoskeletal pain and dermatological infections among child laborers ($p < 0.05$). (4) Conclusions: This research suggests that income measures for households and an education program for both children and parents would expedite the abolition of child labor.

**Keywords:** child labor; working conditions; poverty; schooling; Bangladesh





## 1. Introduction

Child labor is a major social and public health conundrum in many developing countries. Despite increasing research over the last two decades on the detrimental impacts of child labor, it remains a significant concern globally. The International Labor Organization (ILO) has not been able to meet its goal of eradicating child labor (Bourdillon and Carothers 2019), with recent data suggesting over 152 million children are engaged in paid work (International Labor Organization 2017). The prevalence of child labor in low-income countries of Asia and Africa continues to escalate. Asia alone has an estimated 122 million children, aged 5 to 14 years, who are compelled to work for their survival (International Labor Organization 2019). This proliferation is alarming, as it may deprive a section of global children of their rights and put them at risk of physical or mental harm (Wright 2003). The conceptual understanding of child labor varies according to the definitions of the child and their working hours. For example, the Statistical Information and Monitoring Programme on Child Labor (SIMPOC) defined a child laborer as an economically active child under 12 years who works one or more hours per week, an economically active child 14 years or under who works at least 14 h or more per week in activities that are "hazardous by nature or circumstance," and a child 17 years and under who works in an "unconditional worst form of child labor" (trafficked children, children in bondage or forced labor, armed conflict, prostitution, pornography, illicit activities) (Edmonds 2008, p. 3618). According to the Bangladesh Bureau of Statistics (2015), a child between 5–11 years of age working for

any period of time in a non-hazardous job is a child laborer. Children aged 12–17, inclusive, who work for more than 42 h a week in a non-hazardous job are defined as child laborers. The age disparity among the various definitions of child labor is due to differences across geopolitical regions. Research postulates that children usually start working at very young ages and work intensified hours, which further restricts their attendance at school and may also result in malnourishment (Roggero et al. 2007; Ibrahim et al. 2019).

### 1.1. Key Justification of This Study

The predominant reason for conducting this study is that child labor is seen as a public health concern but is an under-researched area. The major hindrance of child labor research is the limited availability of recent and reliable official data to effectively address the nature and extent of child labor. The evidence of adverse health consequences of child labor in the context of developing countries is limited (Ahmed and Ray 2014). A report from the International Labor Organization (International Labor Organization 2018b) stated that legislation cannot eliminate it alone; what is required is effective economic measures that will provide protective action. This statement points to the necessity for more studies on child labor in order to design effective policy measures. In Bangladesh, poverty rates have declined to 9% in 2018. Around 12.9% of the population live below the poverty line and the country is now the 44th largest world economy in nominal terms of GDP (The World Bank 2017). Despite this economic progression, the prevalence of child laborers in Bangladesh is still rampant. Considering child labor in relation to population density, the country is in a most delicate situation in South Asia. In a report from the ILO, it was estimated that among 16.7 million child laborers in South Asia, India accounts for 5.8 million of them, followed by Bangladesh (5 million), Pakistan (3.4 million), and Nepal (2 million) (Khan and Lyon 2015). This suggests that there are other risk factors, and the present research focuses on exploring these factors.

In metropolitan areas of Bangladesh, many child workers are engaged in the informal sector. The informal sector often lies outside the jurisdiction of government legislation (Alam et al. 2015). Given the existing fragile structural systems, limited legal protection, and outdated policies, the capacity to enforce child labor protection laws remains difficult. The debate between child labor and labor standards in Bangladesh has prevailed for several years, with all the players protecting their own interests (Wright 2003). In order to reform and revise existing policies regarding the elimination of child labor, up-to-date research is necessary.

### 1.2. Objectives of the Study

This study was carried out to assess the determinants of child labor and the nature of the child's working environment. This objective was further divided into three research sub-aims, as follows:

(i)　To detect the socio-demographic characteristics of child laborers.
(ii)　To explore the causes of being a child laborer.
(iii)　To analyze the nature of the employment and working environment of child laborers.

## 2. Review of Literature

### 2.1. Child Labor in Bangladesh

The GDP growth rate of Bangladesh has been increasing and peaked at 7.90% in 2019, which is a higher rate than many other developing countries (Trading Economics 2019). Despite moderate advancements in efforts to eliminate child labor, 4.3% (1,326,411) of all children are still child laborers in Bangladesh, with the majority (39.7% and 27.3%) employed in the agriculture and the manufacturing sectors, respectively (Bureau of International Labor Affairs 2018; Bangladesh Bureau of Statistics 2015). The National Child Labor Survey 2013 estimated that there are 3.45 million working children in Bangladesh aged between 5 to 17 years, of which 0.57 million are in urban areas and 0.43 are in city corporation areas (Bangladesh Bureau of Statistics 2015). Studies also observed that child

laborers in Bangladesh work for intensified hours. The Bangladesh Labor Welfare Foundation (2016) revealed that a significant proportion of children in the urban industrial sector toil as long as 16 h a day in environments where there are various workplace hazards. In another study, researchers stated that children in the manufacturing and service sectors of Bangladesh are usually exploited by working on average 43 h per week (Norpoth et al. 2014). This means that around 0.2 million children are employed without attending schools (Bangladesh Bureau of Statistics 2015).

Despite this, the country has already taken significant initiatives to overcome these obstacles. A remarkable advancement in this journey was the ratification of the ILO's Worst Forms of Child Labor Convention (C182) and the UN Convention on the Rights of the Child (Herath and Sharma 2007). Among the concerns at the national level, Bangladesh has been following the British enacted laws to reduce child labor as a British colony, such as the Mines Act, 1923, the Children Act, 1933, The Employment of Children Act, 1938, the Factory Act of 1965, and so forth. After independence, the country enacted the Children Act 1974, currently updated as the Children Act of 2013, the National Children Policy 1994, and the Labor Act of 2006. These initiatives directly contributed to protecting children's rights in Bangladesh (Beaubien 2016; Kalam 2007). Very recently, the Government of Bangladesh also enacted the National Child Labor Elimination Policy (2010). This policy mandates eliminating all forms of child labor from the country by implementing laws of various national acts, providing financial grants, ensuring primary education, strengthening relevant institutional capacity, and raising social awareness (Ministry of Labor and Employment 2010).

### 2.2. Determinants of Child Labor

In order to reduce the extent of child labor in developing countries, extensive studies on the root factors are crucial. Notably, the predominant focus regarding the causal factor of child labor is usually on a single predictor. A plethora of research literature identifies poverty or economic crises as a major cause of child labor (Bourdillon and Carothers 2019; Martin 2013; Fors 2012; Salmon 2005; Basu and Van 1998). However, not all child laborers work due to poverty, rather, in many cases, they work in order to assist in household chores, or on farms as helping hands as part of family ventures. A study conducted by Edmonds (2003) revealed that 92% of working children in Nepal and 87% in Vietnam work on family farms to assist parents, as parents prefer their children to work within their household. Poverty in these families predominantly drives the child into work. The determinants of child labor include financial crisis (Bourdillon and Carothers 2019; Salmon 2005), parental ontogenetic traits such as illiteracy, unemployment conditions (Webbink et al. 2013), household characteristics (resources and structural) (Adonteng-Kissi 2018; Webbink et al. 2013) and the high cost of schooling (Barman 2011). Another methodical study identifies that children who grew up in environments where domestic violence, ruptures in familial relationships, and poor attachments were usual incidents were highly vulnerable to participating in child labor (de Mesquita and de Farias Souza 2018). In addition, the parental debt burden may result in a child becoming a bonded laborer (Basu and Chau 2004), although it is strictly prohibited in many countries. Each of these causes is correlated (Patrinos and Psacharapoulos 1997). Thus, ambiguity often arises in the measurement of top-ranked associated factors. Togunde and Carter (2006) reported that children of parents with high socioeconomic status work fewer hours, which implicitly reflects the influence of income disparity in child labor exploitation. Child labor also increases as gender disparity increases (Ali et al. 2017). For example, Ali et al. (2017) observed that in certain societies, boys are more significantly engaged in child labor than girls because of their biological characteristics and social acceptance. Geography has an impact on children being forced into labor, as the area is poorer. Migration to urban areas is also an underlying cause of child labor in South Asia (Patrinos and Psacharapoulos 1997; Edmonds 2003). Traditional societal norms, cultural beliefs, and customs have also been identified as core contributing factors to child labor (Adonteng-Kissi 2018). Child labor also exists where there are labor-intensive production techniques and industries available.

This suggests that the mode of production practices, the structure of the labor market, and national and international trade arrangements are tacitly responsible for this problem (Fors 2012; Herath and Sharma 2007). The advancement of technology and globalization and the liberalization of international trade further accelerated the rate of child labor in many developing nations (Herath and Sharma 2007). The weakness of state intervention, along with the failure of enforcing labor laws, correspondingly pose impediments to the eradication of child labor (Martin 2013).

*2.3. Hazardous Working Environment*

One of the most gruesome aspects of child labor is the exposure of children to hazardous and unsafe workplace environments, which jeopardize their health and development. Despite this, the concept of hazardous child labor is not properly defined and the consequences of exposure to hazards are still under-reported and undocumented (Ide and Parker 2005). The most commonly identified hazardous sectors for child laborers are mining, construction, and manufacturing, along with small-scale level work in hotels, restaurants, automobile workshops, brick manufacturing, engine vehicle workshops, electric mechanic shops, bakeries, weaving workshops, blacksmith workshops, and so forth. (International Labor Organization 2014; Parker and Overby 2005; Cooper and Rothstein 1995). The ILO estimates 73 million child laborers are engaged in these hazardous occupations (International Labor Organization 2017). Working in hazardous environments exposes children to dangerous substances, agents, or processes, high or low temperatures, high noise levels, and vibrations, damaging their health and well-being. The ILO found common health disorders occur each year, including musculoskeletal damage arising from carrying heavy loads, lung disease from exposure to dusts, and cancers and reproductive disorders due to exposure to pesticides, insecticides, and industrial chemicals. The report also estimated that exposure kills around 22,000 children every year (International Labor Office 2011).

The message of hope beyond all adversity is that a global initiative has been launched on child labor eradication as an important goal of the United Nations (UN) SDG (Sustainable Development Goal) program. The SDG Target 8.7 directly focuses on the elimination of the various types of child labor, such as forced labor, modern slavery, human trafficking, the use of child soldiers, and all other forms of child labor by 2025. Another aspect of this goal is its Target 16.1, which aims to take steps to halt any form of abuse against children (International Labor Organization 2018a, 2018b). As is the case for many nations, Bangladesh is also progressing to fulfill these targets launched by the UN.

Bangladesh is often regarded as a highly vulnerable region, prone to hazardous working environments. A study performed by the London-based Overseas Development Institute found that child laborers in Bangladesh as young as 6 years old work full-time, while others are employed up to 100 to 110 h per week in a variety of hazardous activities, and earn less than USD 2 a day (Beaubien 2016). The UN Committee on the Rights of the Child noted that children in Bangladesh work in five of the worst activities, namely welding, auto workshops, battery recharging, road transport, and tobacco factories (Norpoth et al. 2014). The Bangladesh Labor Force Survey of 2000 reported that a significant proportion of children in Bangladesh, particularly urban child workers, are toiling in hazardous activities, especially in the construction, manufacturing, and domestic sectors, where they work even more than 40 h a week (Salmon 2005). Children working in the mining and agriculture sectors of Bangladesh are more susceptible to the risk of damaging their health and well-being. Children working in the agricultural sector are spreading poisonous fertilizers, pesticides, and herbicides, which is evidently detrimental to their health (Amon et al. 2012). Despite these threats to children in labor, this area occupied derelict attention in child labor research in the context of Bangladesh. Regrettably, Work, Health, and Safety (WHS) induction training for the child or young employees within the workplace is unusual in many developing countries like Bangladesh. No studies were found in the context of

developing countries that indicated that child laborers are provided with education in work, health, and safety provisions.

## 3. Materials and Methods

### 3.1. Study Area

A cross-sectional study was conducted in Sylhet Sadar Upazila (Pathantula, Amberkhana, Shahi Eidgah, Shibgang, and Bondor) of Bangladesh. Households in this city receive the country's second-largest annual remittance from overseas migrant workers per year (BDT 333,125) (Haque et al. 2017). The emerging different types of small-scale businesses and industries in this city often use children as cheap labor. The number of employed people between 10 to 19 years is 31,260 in this city (Bangladesh Bureau of Statistics 2013); this demonstrates the increased prevalence of child workers in this city.

### 3.2. Sample Design and Data Collection

A total of 80 child laborers (between 5 and 17 years) were recruited through two- to three-weeks of repeated visits. As there are no specific data regarding the child labor census (aged 5–17 years) of Sylhet city available on national data portals, the researchers employed a non-probability sampling technique known as snowball sampling. Notably, snowball techniques are useful for identifying hard-to-reach populations (Faugier and Sargeant 1997). A set of piloted unstructured questions were used in the data collection phase. Given that the participants (child laborers) would not be able to understand and fill in the survey themselves because of low literacy levels, a personal interview technique was employed with the identified participants. Each respondent was approached and interviewed separately and assured that all information would remain anonymous and confidential. The data collection proceeded following the approval of the ethics application by the University's Committee for the Human and Ethics Review Board. After obtaining informed consent, data were collected between June 2018 and August 2018.

### 3.3. Data Analysis

The collected data were analyzed, tabulated, and summarized in accordance with the objectives of the study. Of note, the current study administered diverse data analytical devices. The socio-demographic characteristics of participants were studied using simple descriptive statistics. To identify and establish possible determinants of child labor in rank order, this cross-sectional study utilized a newly devised ranking tool, the Influencing Cause's Index (ICI) technique (Ahad et al. 2017). The preliminary ideas were gained from the study conducted by Pandit and Basak (2013) in developing this technique. The pilot-tested questionnaire associated with this technique includes four response level categories ("very high", "high", "little" and "not at all", with weightings of 3, 2, 1, and 0, respectively) for each identified variable. The identification and ordering of possible determinants, applying this tool, followed a two-step process: (i) proportion estimation of response categories (four-point scales) for each variable, and (ii) application of the following equation to compute the ICI value for each determinant:

$$ICI = P_n \times 0 + P_l \times 1 + P_h \times 2 + P_{vh} \times 3$$

where $P_n$ is the percentage of child laborers not having this as an influencing cause, $P_l$ is the percentage of child laborers having this as a little influencing cause, $P_h$ is the percentage of child laborers having this as a high influencing cause, and $P_{vh}$ is the percentage of child laborers having this as a very high influencing cause.

To test the reliability of the above-estimated ICI ranking orders, the Friedman Statistical Test was performed. It provided different mean rank values, which underpinned the admissibility of the ICI technique. The chi-square test was also employed to show the significant differences among the categorical variables of workplace environments. The overall data analysis was conducted using SPSS 26.

## 4. Results

### 4.1. Socio-Demographic Characteristics of Child Laborers

Table 1 indicates the frequency distribution of different socio-demographic attributes of child laborers. The study shows that 61.25% of children aged between 12 and 14 years were engaged in labor, while only 6.25% fall within the age group of 5–7 years. The largest share, 60%, had attended primary level school, whereas 32.5% of the child laborers had never attended school. Approximately 16.25% and 13.75% were employed as rickshaw pullers or in agricultural activities, respectively, and 25% were employed in welding workshops or in retail. In addition, 88.75% came from nuclear families, and in 66.25% of cases, their father was the family head. A large share of child laborers (60%) lived in rental accommodations. Despite households having two other individuals in work, 47.5% of children were forced to seek paid employment.

**Table 1.** Socioeconomic Information of Child Laborers.

| Characteristics | | n (%) |
|---|---|---|
| Age | 5–7 | 5 (6.25) |
| | 8–11 | 13 (16.25) |
| | 12–14 | 49 (61.25) |
| | 15–17 | 13 (16.25) |
| Marital Status | Married | 5 (6.25) |
| | Unmarried | 75 (93.75) |
| Education | Up to Primary | 48 (60) |
| | Up to Secondary | 6 (7.5) |
| | No Educational Qualification | 26 (32.5) |
| Occupation | Agriculture | 11 (13.75) |
| | Electronic/Mechanic Worker | 8 (10) |
| | Welding Worker | 10 (12.5) |
| | Hotel/Restaurant Worker | 5 (6.25) |
| | Tempo Helper | 5 (6.25) |
| | Automobile Helper | 7 (8.75) |
| | Salesperson | 10 (12.5) |
| | Rickshaw Puller | 13 (16.25) |
| | Construction Worker | 8 (10) |
| | Others | 3 (3.75) |
| Family Size | Nuclear | 71 (88.75) |
| | Extended | 9 (11.25) |
| Family Head | Father | 53 (66.25) |
| | Mother | 20 (25) |
| | Brother | 7 (8.75) |
| | Sister | 0 (0) |
| Housing Pattern | Own house | 27 (33.75) |
| | Rented house | 48 (60) |
| | No house | 5 (6.25) |
| Earning Member of Family | One | 6 (7.5) |
| | Two | 38 (47.5) |
| | Three | 28 (35) |
| | More than three | 8 (10) |
| Positive Relationship with Parents | Yes | 74 (92.5) |
| | No | 6 (7.5) |
| Father's Educational Qualification | Primary | 19 (23.75) |
| | Secondary | 4 (5) |
| | No educational qualification | 57 (71.25) |

**Table 1.** *Cont.*

| Characteristics | | n (%) |
|---|---|---|
| Mother's Educational Qualification | Primary | 16 (20) |
| | Secondary | 2 (2.5) |
| | No educational qualification | 62 (77.5) |
| Father's Occupation | Day Laborer | 16 (20.00) |
| | Rickshaw Puller | 9 (11.25) |
| | Fisherman | 3 (3.75) |
| | Business | 10 (12.50) |
| | Agriculture | 12 (15) |
| | Carpenter | 3 (3.75) |
| | Guard men | 2 (2.50) |
| | Welding Worker | 3 (3.75) |
| | Tempo Helper | 2 (2.50) |
| | Driver | 6 (7.50) |
| | Others | 3 (3.75) |
| | Unemployed | 11 (13.75) |
| Mother's Occupation | Only Housewife | 53 (66.25) |
| | Domestic Laborer | 18 (22.5) |
| | Others | 9 (11.25) |

This cross-sectional study revealed that the fathers of 57 respondents and the mothers of 62 respondents had no formal educational qualifications. Approximately 20%, 15%, and 12.75% of child laborers disclosed that their fathers were engaged in day labor, agriculture, and small-scale businesses, respectively, while 13.75% were unemployed. Up to 66.25% of the mothers of child laborers were housewives.

### 4.2. Causes of Being a Child Laborer

The preliminarily identified determinants of child labor are poverty, parental unemployment, low aspirations of parents, uneducated family members, trouble at home, parental debt, the high cost of education, poor schooling opportunities, the high demand for unskilled and cheap labor, urban migration, natural calamity, and early marriage. These identified characteristics were determined from previous research in order of importance. An ICI measure was used to detect their ranking order as a cause of child labor.

Table 2 illustrates the mean value of four response categories (No, Little, High, and Very High) for each influencing cause of child labor. For instance, in the case of "Poverty", the mean value for the "No" response category was 0%, followed by 2.5% for "Little", 27.5% for "High", and 70.0% for "Very high" influencing causes of child labor. In this manner, the percentages (%) for the remaining categories for each variable were computed. The chi-square test statistics revealed that there were significant differences among the response categories for every possible determinant of child labor ($p < 0.01$), with an exception for categories involved in the "trouble at home".

**Table 2.** Categorization of the responses according to the roots of being child laborers.

| Item | Response Categories | Percentage of Responses |
|---|---|---|
| Poverty | No | 0.00 |
| | Little | 2.5 |
| | High | 27.5 |
| | Very high | 70.0 |
| Unemployment condition of family members | No | 1.3 |
| | Little | 15.0 |
| | High | 73.8 |
| | Very high | 10.0 |

**Table 2.** *Cont.*

| Item | Response Categories | Percentage of Responses |
|---|---|---|
| Low aspirations of parents | No | 10.0 |
| | Little | 41.3 |
| | High | 33.8 |
| | Very high | 15.0 |
| Uneducated family members | No | 1.3 |
| | Little | 7.5 |
| | High | 57.5 |
| | Very high | 33.8 |
| Trouble at home | No | 23.8 |
| | Little | 33.8 |
| | High | 28.7 |
| | Very high | 13.8 |
| Parents under heavy debt | No | 50.0 |
| | Little | 13.8 |
| | High | 15.0 |
| | Very high | 21.3 |
| High cost of education | No | 2.5 |
| | Little | 5.0 |
| | High | 32.5 |
| | Very high | 60.0 |
| Poor schooling opportunity | No | 1.3 |
| | Little | 7.5 |
| | High | 48.8 |
| | Very high | 42.5 |
| Huge demand of unskilled and cheap labor | No | 3.8 |
| | Little | 31.3 |
| | High | 47.5 |
| | Very high | 17.5 |
| Urban migration | No | 32.5 |
| | Little | 2.5 |
| | High | 36.3 |
| | Very high | 28.7 |
| Natural calamity | No | 81.3 |
| | Little | 5.0 |
| | High | 7.5 |
| | Very high | 6.3 |
| Early marriage | No | 95.0 |
| | Little | 5.0 |
| | High | 0.00 |
| | Very high | 0.00 |
| Others | No | 88.8 |
| | Little | 10.0 |
| | High | 1.3 |
| | Very high | 0.00 |

Table 3 indicates the calculated rank values for the 13 identified causes of child labor by using the ICI equation. Table 3 illustrates that "Poverty" is ranked first (ICI Score: 267.50) among the 13 recognized causes based on ICI values. "High cost of education" is ranked in the second order in the ICI ranking measure (ICI score: 250). Correspondingly, "Poor schooling opportunity" is positioned in the third-ranking order, with an ICI score of 232.6. "Uneducated family members", "Unemployment condition of family members", "High demand of unskilled and cheap labor", "Urban migration", "Low aspiration of parents", "Trouble/conflicts at home", "Parents under heavy debt", "Natural calamity", "Others",

and "Early marriage" were ranked in the 4th, 5th, 6th, 7th, 8th, 9th, 10th, 11th, 12th, and 13th orders, respectively, as influencing causes of child labor. The Friedman Test shows the parallel mean rank order of child labor indicators is similar to the ICI measure, which reflects the reliability of the ICI measure. Of note, the Friedman chi-square test shows that there is an overall significant difference among the mean ranks of related variables ($p < 0.001$).

**Table 3.** Influencing Cause's Index (ICI) and Friedman Test Rank order of 13 selected causes faced by the child laborers.

| Item | ICI Score | ICI Rank | Friedman Test (Mean Rank) |
|---|---|---|---|
| Poverty | 267.50 | 1 | 10.84 |
| Unemployment condition of family members | 192.60 | 5 | 8.11 |
| Low aspirations of parents | 153.9 | 8 | 7.11 |
| Uneducated family members | 223.9 | 4 | 9.23 |
| Trouble at home | 132.6 | 9 | 6.28 |
| Parents under heavy debt | 107.7 | 10 | 5.78 |
| High cost of education | 250.0 | 2 | 10.24 |
| Poor schooling opportunity | 232.6 | 3 | 9.53 |
| Huge demand of unskilled and cheap labor | 178.4 | 6 | 7.66 |
| Urban migration | 161.2 | 7 | 7.38 |
| Natural calamity | 38.9 | 11 | 3.50 |
| Early marriage | 5.00 | 13 | 2.58 |
| Others | 12.6 | 12 | 2.77 |

$p < 0.001$ (Friedman Test).

### 4.3. Nature of Employment and Working Environment

Table 4 demonstrates the contemporary characteristics of employment and the working environment that would increase the vulnerability conditions of child laborers by impairing their physical, psycho-social, and cognitive development. This cross-sectional study explored certain core characteristics that children usually experience in their workplaces. The findings revealed that among the participants, 73.75% of the child laborers were in paid employment, while 3.75% were unpaid workers ($p < 0.05$). A total of 72.5% were working more than 8 h a day, which violates the labor laws, while 5% spent 7 h at work, and 15% of children worked less than 7 h a day ($p < 0.05$). A total of 46.25% of child laborers were required to walk to their workplaces, while only 13.75% had the opportunity to use buses ($p < 0.05$). Approximately, 88.7% revealed that they were not paid for additional duties, except 11.25% of child laborers ($p < 0.05$).

The majority (58.75%) of the child laborers were paid a daily wage, followed by 3.75% and 37.5% who were paid on a weekly and monthly basis, respectively ($p < 0.05$). There are no significant differences found among the numbers of child labourer who were spending different leisure times at workplaces ($p > 0.05$). In addition, the study shows that a total of 38.75% of the children had no leave provisions in the work arrangements, while 61.25% did ($p < 0.05$). Approximately 90% of the child laborers had no training facilities at their workplaces, although around 67.50% reported that they were supported by their colleagues at the workplace when needed ($p < 0.05$). Notably, the study revealed no differences in the outcomes of receiving hygienic sanitation facilities at workplaces. The majority (35%) of the child laborers had experienced physical pain due to their workload, 23.75% complained of digestive disorders, and 18.75% had dermatological infections ($p < 0.05$). Despite these health adversities, only 25% of the child laborers received medical assistance from their workplaces, whereas 75% were deprived of these facilities ($p < 0.05$). Child laborers were indifferent regarding the possibility of gaining services, including safety or security measures, at their current workplaces ($p > 0.05$). The confidence interval was also estimated to give a precise range of values for each characteristic of employment and working environment, within which the true population mean lies (a 95% chance).

**Table 4.** Nature of Employment and Working Environment.

| Items | | n (%) | CI (at 95%) | *p* Value |
|---|---|---|---|---|
| Nature of Employment | Unpaid workers | 3 (3.75) | [0.8, 10.6] | |
| | Paid workers | 59 (73.75) | [62.7, 83.0] | 0.000 ** |
| | Self-employed | 18 (22.5) | [13.9, 33.2] | |
| Working Span (Hours/day) | Less than 7 h | 12 (15.00) | [8.0, 24.7] | |
| | 7 h | 4 (5.00) | [1.4, 12.3] | |
| | 8 h | 6 (7.5) | [2.8, 15.6] | 0.000 ** |
| | More than 8 h | 58 (72.5) | [61.4, 81.9] | |
| Mode of Transportation | By walking | 37 (46.25) | [35.0, 57.8] | |
| | By bus | 11 (13.75) | [7.1, 23.3] | |
| | By bicycle | 17 (21.25) | [12.9, 31.8] | 0.000 ** |
| | Other means | 15 (18.75) | [10.9, 29.0] | |
| Overtime Working Facility | Yes | 6 (7.5) | [2.8, 15.6] | |
| | No | 74 (92.5) | [84.4, 97.2] | 0.000 ** |
| Payment for Additional Duty | Yes | 9 (11.25) | [5.3, 20.3] | |
| | No | 71 (88.7) | [79.7, 94.7] | 0.000 ** |
| Method of Payment | Daily | 47 (58.75) | [47.2, 69.6] | |
| | Weekly | 3 (3.75) | [0.8, 10.6] | 0.000 ** |
| | Monthly | 31 (37.5) | [26.9, 49.0] | |
| Leisure Hour per Working Day | Up to 15 min | 14 (17.50) | [9.9, 27.6] | |
| | 15 to 30 min | 24 (30.00) | [20.3, 41.3] | |
| | 30 to 60 min | 22 (27.50) | [18.1, 38.6] | 0.423 |
| | More than 1 h | 20 (25.00) | [16.0, 35.9] | |
| Proper Leave Facilities | Yes | 49 (61.25) | [49.7, 71.9] | |
| | No | 31 (38.75) | [28.1, 50.3] | 0.044 * |
| Training Facilities by Current Workplace | Yes | 8 (10.00) | [3.6, 17.2] | |
| | No | 72 (90.00) | [82.8, 96.4] | 0.000 ** |
| Support from Other Employees during Work | Yes | 54 (67.50) | [56.1, 77.6] | |
| | No | 26 (32.50) | [22.4, 43.9] | 0.002 ** |
| Hygienic Sanitation Facilities at Workplace | Yes | 41 (51.25) | [39.8, 62.6] | |
| | No | 39 (48.75) | [37.4, 60.2] | 0.823 |
| Affected by Common Diseases Due to Work | Cough and Cold | 13 (16.25) | [8.9, 26.2] | |
| | Body Pain | 28 (35.00) | [24.7, 46.5] | |
| | Digestive Problems | 19 (23.75) | [14.9, 34.6] | 0.001 ** |
| | Dermatological Infections | 15 (18.75) | [10.9, 29.0] | |
| | Headache | 5 (6.25) | [2.1, 14.0] | |
| Medical/Health Support | Yes | 20 (25.00) | [23.9, 26.1] | |
| | No | 60 (75.00) | [58.9, 61.1] | 0.01 ** |
| Safety Measure at Workplace | Yes | 33 (41.25) | [30.4, 52.8] | |
| | No | 47 (58.75) | [47.2, 69.6] | 0.118 |

Statistically significant at level of * $p < 0.05$, ** $p < 0.01$.

## 5. Discussion

This paper contributes to the existing research by evaluating the possible determinants of child labor and the patterns of the working environment from a sample of 80 child laborers in Bangladesh.

Data revealed that most of the children are between 12 to 14 years old and not attending school. Most children are employed in rickshaw pulling, sales jobs, or farming, which confirms a study conducted by Rahman et al. (1999). Most of the parents of these child laborers had no formal education and were unemployed. Employed parents were predominantly found to be farmers, small-scale businessmen, and miscellaneous day laborers. A plethora of research findings also indicates that a substandard socio-

demographic status often leads children to the workforce, which is consistent with the findings of this study (Barman 2011; Togunde and Carter 2006). This study found that almost all the child laborers had positive relationships with their parents.

The present study undertook an assessment of the factors responsible for child labor. Notably, the theoretical or empirical research on this topic has received much attention in recent years (Fors 2012). The existing research evidence suggests that poverty, illiteracy (Bourdillon and Carothers 2019), household size, cultural values (Adonteng-Kissi 2018), adult unemployment, higher schooling cost (Barman 2011), and credit market constraint (Adonteng-Kissi 2018) are the core causes of child labor. Based on this growing body of evidence and systematic survey, the present study identifies a list of possible determinants of child labor, applying a rank-order technique. The Influencing Causes Index (ICI) identified poverty as the top-ranked factor and the most powerful force driving children into labor. A major perplexing outcome of this evaluation is that participants nominated "the high cost of education" as the second-ranked factor of being in child labor, despite the fact that primary education is free for all children through the Compulsory Primary Education Act 1993 in Bangladesh (Rabbi 2018). The study identifies that parental illiteracy or low levels of education of family members also has a substantial adverse impact on child labor. Children are less likely to work and more likely to attend the school where the family head has received some education (Ali et al. 2017). Parental education level is a factor in a child engaging in the workplace. The fourth-ranked factor identified was "poor schooling opportunity". These three factors identify education as a key to deprivation. Education is an important factor in eliminating child labor. Additionally, this new-flagged distinctive ranking technique demonstrated other underlying persuasive indicators, such as the unemployment condition of family members, demand for unskilled and cheap labor, and urban migration.

The complex nature of employment and unsafe working environments adversely affect children. The present study explored the core domains of these trends. Though the ILO Global statistics reported that most child laborers are engaged in unpaid household services (International Labor Organization 2017), the findings here reveal that the majority of the child laborers who work in urban areas are paid. In addition, the study shows that the majority (72.5%) of child laborers work over eight hours a day, which breaches the international labor laws (International Labor Organization 2005). Child laborers are not paid for overtime despite working for low wages. The most damning issue is that only a quarter of the children get more than a one-hour break from work.

The Child Labor Act in Bangladesh provides for safe and healthy work environments for children (Dey 2008), but it would appear that the regulations are not being enforced. In addition, a high proportion of the child laborers have inadequate leave opportunities, poor training, little access to healthy sanitation facilities, or medical or health support. This study uncovered that the majority of participants do not have these at their workplaces. It is obvious that the incidence of health risk was exacerbated by the lack of mandatory safety measures, including workplace safety training, providing personal protective equipment or workplace protective equipment, such as labeling hazardous zones and fire emergency services. A study conducted by Ibrahim et al. (2019) revealed the leading negative effects on the health of child laborers were musculoskeletal injuries, HIV infections, and other infectious diseases. It is worth noting that a high prevalence of physical illness, including pain (35%), stomach disorders (23.75%), and dermatological infections (18.75%), were observed in this study among the participants, although medical evidence was not available. These injury and ill-health rates are the results of working conditions, along with the developmental and physical impacts on young children toiling for long hours. The incidence of health risk could be reduced by implementing mandatory safety measures, including the wearing of protective equipment, mechanical aids where possible, labeling hazardous zones, first aid services, and flood or fire emergency services (Cooper and Rothstein 1995; Bourdillon and Carothers 2019).

The policy implication of this study covers several areas. Firstly, public policies need to encourage disadvantaged children to enroll in formal primary and secondary education in metropolitan areas. The census report of 2011 revealed that still around one-fifth of children in Bangladesh are deprived of formal education (Bangladesh Bureau of Statistics and United Nations International Children's Emergency Fund 2014). This needs to be taken up by related institutional bodies. Secondly, an effective instrument to generate income support measures for poor households should be developed to curb the prevalence of child labor, especially where access to industry or capital markets is limited. It is already mandated by the ILO that to achieve the eradication of child labor by 2025, every nation needs to promote decent working conditions for adults of poor households (International Labor Organization 2018b). To keep pace with this global drive, Bangladesh also needs to focus on this target. The final approach suggested by the current study is to devise new or revised policy interventions focusing on adult education. The research postulates that adult learning and education increases the opportunity to escape poverty and inequalities. Since the 1990s, Bangladesh has also achieved relatively high successes in adult education but has not fulfilled the Education for all (EFA) goal (Ahmed 2009). The goal sets a limit of achieving a 50% improvement in adult literacy along with other criteria (World Conference on Education for All: Meeting Basic Learning Needs 1990).

## 6. Conclusions

Child labor is a major problem across the developing world and violates child rights. This study highlights the major determinants of child labor in one area of Bangladesh highlighting the impact on the health and development of these children. It points to the failures of various policy and legal protections put in place for these children and the problem of low levels of education for parents. Government intervention is required to ensure families see the value of education for their children in breaking the cycle of poverty. Similarly, there is an urgent need for workplace safety legislation to be enforced along with welfare provisions for the unemployed to encourage schooling for children over seeking work.

**Author Contributions:** The preparation of manuscripts including designing the study, literature search, statistical analysis, data collection, and analyses of the data were accomplished by M.A.A. The writing of the first draft of the manuscript was completed by M.A.A. Y.K.P. contributed to the statistical data and analysis, E.W., provided major input into the writing of the various drafts, and M.A.A. and M.C. oversaw the conduct of the project, organized the ethics, and were part of the initial research design. All authors have read and agreed to the published version of the manuscript.

**Funding:** This work was supported by funds provided by the University Grant Commission (UGC), Dhaka, Bangladesh.

**Institutional Review Board Statement:** The study was conducted according to the guidelines of the Declaration of Helsinki, and approved by the Institutional Review Board (or Ethics Committee) of Sylhet Agricultural University (protocol code #ARP2017001 and date of approval is 15 August 2017).

**Informed Consent Statement:** Informed consent was obtained from all subjects involved in the study.

**Acknowledgments:** The authors are grateful to the children of Sylhet city corporation, who participated in the interview.

**Conflicts of Interest:** The authors declare no conflict of interest.

**Ethics Approval:** The ethics approval for the collection of data was obtained from the University Research Ethics Committee.

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
