# Peer review of "Urban Child Labor in Bangladesh: Determinants and Its Possible Impacts on Health and Education"

_socsci, doi:10.3390/socsci10030107_

Round 1
Reviewer 1 Report
Reviewer report
Dear Author(s):
Synopsis of the review This is a good paper which, however, has some flaws which, in my view, do not recommend the publication of the paper in its present state. Once these are corrected, I would be very happy for the paper to be accepted and, consequently, published.
These are as follows: - Structuring of the paper. I consider that the introduction is extremely long and should be reworked, with a brief mention of the state of the literature, which is quite copious on the issue of Child Labour. I therefore suggest the following: Keep lines 21-43 in the introduction section. Immediately after these lines add the lines After these lines 119-148 (Justification+objectives). In this way a prospective reader of the article will get a more accurate picture of the contributions of your work, objectives and justification. Please avoid unnecessary line-breaks as in lines [146-18] of the current version of the paper. Please include the remaining lines of the introduction (44-111) in a subsection called (State of the Art or literature review).
I suggest that in this new subsection you include the following bibliographical references: Nogler, L.; Pertile, M. & Nesi, G. (2016), Child Labour in a Globalized World: A Legal Analysis of ILO Action, Taylor & Francis. Sharma, K. & Herath, G. (2016), Child Labour in South Asia, Taylor & Francis. Sālāhauddina, K. & Association, B. W. W. (2001), Child labour in Bangladesh: the early years, Palok Publishers & Bangladesh Women Writers Association. Khair, S.; Office, I. L. & on the Elimination of Child Labour, I. P. (2005), Child Labour in Bangladesh: A Forward Looking Policy Study, International Labour Office, International Programme on the Elimination of Child Labour (IPEC).
In this sense, it will contextualise your study much better, emphasising that it is also due to the process of globalisation, its rootedness in South Asia, as well as the phenomenology of this process in Bangladesh. - Please mention one very important aspect: the eradication of child labour is one of the targets set by the 2030 Agenda. -
The non-inclusion of figures in your work does not in any way penalise its quality, as well as the methodology and results used, but, in any case, I suggest that you include in your work a graph showing the evolution of child labour in Bangladesh in recent times, and even better if in this graph the evolution of the world total is also represented. - Unfortunately, I consider the conclusions to be too limited.
Please note that the results give rise to a final discussion that is much longer than only 8 lines. Also, please note that your final comments could be made about Bangladesh as well as about any other nation that continues to experience the sad phenomenon of child labour: Please elaborate further on this part and indicate in the light of the literature what would be the differential facts of child labour in Bangladesh compared to other countries in the same geopolitical area.
Best regards,
The reviewer.
Reviewer 2 Report
Reviewer’s comments on manuscript on Child Labour in Bangladesh
For the reviewer this is a challenging article. The subject matter is of crucial importance, and the data collected, if properly presented, could add to our ongoing understanding of the problem of child labour in Bangladesh.
However, the article needs to be reorganised, and rewritten before it can be properly evaluated. I will be glad to give an opinion on future drafts.
First: The review of previous literature is inadequate. The author(s) have, inexplicably, not referred to or analysed many relevant studies. This must be done for an article to have scholarly merit. I attach a list of relevant studies. Most of these have links to free PDF downloads on Google Scholar.
Second: Results are presented before any statement of the research population and research methods. This not an acceptable form of academic presentation. The section on the population studied, and research methods used must precede the presentation of results. (The use of the snowball sampling method of locating a difficult-to-reach population is to be commended). Ultimately, this research would have worked better as a qualitative study.
Third: Since only the located population was studied, I am extremely puzzled as to how Chi-squared, rank order analysis, or significance testing could have been attempted when there is no control or contrast group. For me, Ahad’s causal analysis just doesn’t work. This article can only survive if it has (a) a good, comprehensive analysis of previous studies; (b) a description of current conditions and background factors for a located population of child labourers; (c) consideration of social policy solutions, including the economic and social forces which continue to drive the demand for child labour.
I suggest that Section 4.3 be deleted completely, and the puzzling and ultimately meaningless P values be deleted from the Tables.
The following references should be consulted, analysed and cited (this list is not exhaustive):
Amin, S., Quayes, M. S., & Rives, J. M. (2004). Poverty and other determinants of child labor in Bangladesh. Southern Economic Journal, 876-892.
Salmon, C. (2005). Child labor in Bangladesh: Are children the last economic resource of the household?. Journal of Developing Societies, 21(1-2), 33-54.
Kuddus, A., & Rahman, A. (2015). Human Right Abuse: A Case Study on Child Labor in Bangladesh. International Journal of Management and Humanities, 1(8), 1-4.
Islam, A., & Choe, C. (2013). Child labor and schooling responses to access to microcredit in rural Bangladesh. Economic Inquiry, 51(1), 46-61.
Shafiq, M. N. (2007). Household schooling and child labor decisions in rural Bangladesh. Journal of Asian Economics, 18(6), 946-966.
Ruwanpura, K. N., & Roncolato, L. (2006). Child rights: An enabling or disabling right? The nexus between child labor and poverty in Bangladesh. Journal of Developing Societies, 22(4), 359-378.
Hosen, M., Khandoker, S., & Islam, S. M. (2010). Child labor and child education in Bangladesh: Issues, consequences and involvements. International Business Research, 3(2).
Nielsen, M. E. (2005). The politics of corporate responsibility and child labour in the Bangladeshi garment industry. International Affairs, 81(3), 559-580.
Alam, S., Mondal, N. I., & Rahman, M. (2008). Child labor due to poverty: A study on Dinajpur district, Bangladesh. The social sciences, 3(5), 388-391.
Kamruzzaman, M. (2015). Child victimization at working places in Bangladesh. American Journal of Applied Psychology, 4(6), 146-159.
Khanam, R. (2008). Child labour and school attendance: evidence from Bangladesh. International Journal of Social Economics.
Kamruzzaman, M., & Hakim, M. A. (2018). A review on child Labour criticism in Bangladesh: An Analysis. International Journal of Sports Science and Physical Education, 3(1), 1-8.
Uddin, M. N., Hamiduzzaman, M., & Gunter, B. G. (2009). Physical and psychological implications of risky child labor: A study in Sylhet city, Bangladesh.
Bazen, S., & Salmon, C. (2010). The impact of parental health on child labor: the case of Bangladesh. Economics Bulletin, 30(4), 2549-2557.
Shahjahan, M. B. (2016). Protecting Child Labor in Bangladesh under Domestic Laws. Open Access Library Journal, 3(04), 1.
Amin, S., Quayes, S., & Rives, J. M. (2006). Market work and household work as deterrents to schooling in Bangladesh. World Development, 34(7), 1271-1286.
Putnick, D. L., & Bornstein, M. H. (2015). Is child labor a barrier to school enrollment in low-and middle-income countries?. International journal of educational development, 41, 112-120.
Round 2
Reviewer 1 Report
Dear author(s),
I consider that you have carried out the improvement proposals suggested by me (more or less), so in my opinion, this work should be accepted in its present state.
Best,
The reviewer.
Author Response
Thank you for your comments.
Reviewer 2 Report
This revision shows much improvement, and the paper is now suitable for publication in an academic journal.
There remain some minor English language errors which need to be checked.
Author Response
Thank you for your comments.